# Artificial Intelligence Applications for Osteoporosis Classification Using Computed Tomography

**DOI:** 10.3390/bioengineering10121364

**Published:** 2023-11-27

**Authors:** Wilson Ong, Ren Wei Liu, Andrew Makmur, Xi Zhen Low, Weizhong Jonathan Sng, Jiong Hao Tan, Naresh Kumar, James Thomas Patrick Decourcy Hallinan

**Affiliations:** 1Department of Diagnostic Imaging, National University Hospital, 5 Lower Kent Ridge Rd, Singapore 119074, Singaporeandrew_makmur@nuhs.edu.sg (A.M.); xi_zhen_low@nuhs.edu.sg (X.Z.L.); jonathan_wz_sng@nuhs.edu.sg (W.J.S.); james_hallinan@nuhs.edu.sg (J.T.P.D.H.); 2Department of Diagnostic Radiology, Yong Loo Lin School of Medicine, National University of Singapore, 10 Medical Drive, Singapore 117597, Singapore; 3University Spine Centre, Department of Orthopaedic Surgery, National University Health System, 1E Lower Kent Ridge Road, Singapore 119228, Singapore; jonathan_jh_tan@nuhs.edu.sg (J.H.T.); dosksn@nus.edu.sg (N.K.)

**Keywords:** artificial intelligence, machine learning, deep learning, osteoporosis, imaging, computed tomography

## Abstract

Osteoporosis, marked by low bone mineral density (BMD) and a high fracture risk, is a major health issue. Recent progress in medical imaging, especially CT scans, offers new ways of diagnosing and assessing osteoporosis. This review examines the use of AI analysis of CT scans to stratify BMD and diagnose osteoporosis. By summarizing the relevant studies, we aimed to assess the effectiveness, constraints, and potential impact of AI-based osteoporosis classification (severity) via CT. A systematic search of electronic databases (PubMed, MEDLINE, Web of Science, ClinicalTrials.gov) was conducted according to the Preferred Reporting Items for Systematic Reviews and Meta-Analyses (PRISMA) guidelines. A total of 39 articles were retrieved from the databases, and the key findings were compiled and summarized, including the regions analyzed, the type of CT imaging, and their efficacy in predicting BMD compared with conventional DXA studies. Important considerations and limitations are also discussed. The overall reported accuracy, sensitivity, and specificity of AI in classifying osteoporosis using CT images ranged from 61.8% to 99.4%, 41.0% to 100.0%, and 31.0% to 100.0% respectively, with areas under the curve (AUCs) ranging from 0.582 to 0.994. While additional research is necessary to validate the clinical efficacy and reproducibility of these AI tools before incorporating them into routine clinical practice, these studies demonstrate the promising potential of using CT to opportunistically predict and classify osteoporosis without the need for DEXA.

## 1. Introduction

Osteoporosis is characterized by low bone mineral density (BMD) and microstructural degradation of the bone tissue [1], rendering bones more brittle and susceptible to fractures. According to a recent meta-analysis conducted in 2021 by Salari et al. [2], the global prevalence of osteoporosis in women was 23.1%, while the prevalence of osteoporosis among men was 11.7%, with the prevalence varying greatly between different countries [3]. The main complication of osteoporosis is fragility fractures, which are frequently linked to heightened mortality and morbidity [4,5,6]. Substantial physical, psychological, social, and economic repercussions due to significant osteoporotic fractures have been extensively documented in prior studies [7,8,9,10]. These include reduced quality of life [11], increased healthcare costs [12,13], increased mortality risks, limited physical activity, and loss of independence [14,15]. As a result, the early diagnosis of osteoporosis is crucial for timely intervention and the prevention of osteoporotic fractures and their complications [16,17,18].

Dual-energy X-ray absorptiometry (DEXA) is endorsed by the World Health Organization (WHO) as the gold standard for evaluating BMD and diagnosing osteoporosis [19,20,21]. The BMD values derived from DEXA are converted into T-scores, which are calculated on the basis of the difference between the individual’s BMD and reference population mean divided by the standard deviation of the population mean [22]. A T-score of −2.5 or less indicates osteoporosis, and a T-score between −1.0 and −2.5 is defined as osteopenia [23,24].

While DEXA remains the most commonly utilized quantitative radiologic method for assessing bone mass [25,26] due to its non-invasiveness and cost-efficiency [27,28,29], certain limitations must be considered. Firstly, DEXA’s diagnostic capability is mainly confined to BMD alone [30,31] and suboptimal screening rates have been reported [32,33]. Additionally, it is constrained by its planar technique (two-dimensional measurement) for assessing and quantifying BMD and predicting fracture risk [34,35]. Moreover, DEXA measurements are sensitive to degenerative changes, leading to the potential overestimation of BMD [36,37,38], and the presence of overlying structures (such as atherosclerosis [39]) or morphological abnormalities (post-laminectomy, metallic implants, etc. [40,41,42]) may also impact BMD measurements.

Dual-energy computed tomography (DECT) is a radiological technique that can be used for measuring bone mineral density (BMD). The concept of using dual-energy computed tomography (DECT) to evaluate BMD was first studied more than four decades ago [43]. DECT measures the attenuation of X-rays as they pass through the bone. This attenuation is affected by the density of the tissue, and by acquiring images at different energy levels (typically high and low energy levels), DECT can calculate the BMD of the bone in the region of interest [44,45].

DECT has shown similar sensitivity to the gold standard DEXA for the evaluation of bone density and the prediction of associated osteoporotic fractures [46,47,48]. For instance, Booz et al. [45] reported a DECT sensitivity of up to 96.0% and a specificity of 93.0% in detecting osteoporosis compared with DEXA. Another study by Gruenewald et al. [49] found that DECT-derived BMD exhibited a sensitivity of 85.5% and a specificity of 89.2% in predicting osteoporotic-associated fractures. In addition to its diagnostic accuracy, DECT offers distinct advantages, including its capability to assess extended dimensional information [50] and to evaluate local changes in BMD. Notably, DECT excels in accurately distinguishing between trabecular and cortical bone, providing valuable insights into the trabecular bone microstructure, such as local parameters like trabecular spacing and connectivity. These capabilities are crucial for the comprehensive evaluation of bone health [51,52]. However, DECT has not been widely used to screen for osteoporosis due to the high radiation dose [53,54], which is a limitation of its unique diagnostic capabilities [55]. Furthermore, it often requires an in-scan calibration phantom [56,57,58], which makes it difficult to use it for routine BMD measurements in CT scans acquired for indications other than BMD measurements [59] or the retrospective measurement of BMD [60,61].

One potential future innovation is to evaluate bone mineral density (BMD) exclusively through CT scans, eliminating the need for DEXA or DECT scans. This could have significant clinical implications for several reasons: first, it allows for the simultaneous provision of both anatomical visualization and quantitative data [62,63]; second, individuals who have undergone CT scans as part of routine health assessments or other medical indications could be screened automatically [64,65,66]; third, a vast CT database could be leveraged to identify patients that may require referral and treatment [67,68,69]; and finally, this approach could lead to decreased expenses [70] and radiation exposure [71,72,73], as the patient may not need to undergo further radiological investigations before the diagnosis and treatment of osteoporosis.

BMD values extracted from CT images were initially determined by establishing a positive correlation between attenuation values or radiodensity (expressed as Hounsfield Units (HUs)) measured at various locations on CT scans and corresponding BMD values obtained from DEXA dating back as early as 2013 [74]. Subsequently, many other studies [67,75,76,77,78] confirmed the feasibility of trabecular HU values for screening osteoporosis on CT. However, most studies have involved manual or semi-automatic segmentation, which is often time-consuming due to the required manual input and/or computing power [79] and other limitations [80,81,82].

With the recent advent of artificial intelligence (AI) technology, machine learning and deep learning models have applications in osteoporosis detection and classification. These applications encompass osteoporosis risk prediction [83,84,85] and fracture risk assessment [86,87,88,89]. Notably, AI has been employed to streamline the complex processing of CT images and enhance automated segmentation [90,91,92], which has been utilized for BMD measurement and classification in multiple studies [93,94,95,96,97,98,99,100,101,102,103]. Radiomics involves the extraction and analysis of a large number of quantitative features from medical images, such as CT scans. These features capture intricate details in the images (beyond the human eye), enabling a more comprehensive understanding of the underlying tissue characteristics. Radiomics and texture analysis with deep learning techniques have also been used to analyze trabecular bone structure in CT images, providing insights into bone quality and microstructure that are not attainable through DEXA or visual measurements alone, thereby improving the accuracy of diagnosing osteoporosis [104,105]. While many studies have focused on the use of AI for classifying or detecting osteoporosis, they often suffer from limitations, such as single-center designs, limited patient samples, and a lack of validation in real clinical settings.

While several studies in the literature have examined the use of AI to classify osteoporosis in CT images, these studies exhibit significant variation and heterogeneity. A literature gap exists because no studies have systematically consolidated and synthesized these varied research efforts for a comprehensive analysis and summary. Hence, this review article aims to provide an overview of the available evidence on the effectiveness and value of AI techniques in diagnosing osteoporosis and classifying BMD using CT imaging. In this study, our classification of BMD was defined as the severity of osteoporosis, specifically distinguishing between normal BMD and low BMD (including osteopenia and osteoporosis) on the basis of the WHO definition derived from the T-score.

## 2. Materials and Methods

### 2.1. Literature Search Strategy

A systematic search of the major electronic databases (PubMed, MEDLINE, Web of Science, and ClinicalTrials.gov) was conducted in concordance with the Preferred Reporting Items for Systematic Reviews and Meta-Analyses (PRISMA) guidelines using keywords and/or medical subject headings (MeSH) for the following key terms: (“Artificial intelligence” OR “AI” OR “deep learning” OR “machine learning” OR “convolutional neural network*” OR “neural network” OR “radiomics”) AND (“osteoporosis” OR “osteopenia” OR “osteopaenia” OR (“bone” AND “mineral” AND “density”) OR “BMD”) AND (“DEXA” OR “absorptiometry”) AND (“CT” OR (“Computed” AND “Tomography”)). Two authors (W.O. and R.L.) performed independent reviews of the collected references and selected the appropriate studies for detailed full-text screening. The last date of the reference and literature search was 14 August 2023. Any potential conflicts were resolved by consensus or by appeal to a third author (J.T.P.D.H).

### 2.2. Study Screening and Selection Criteria

No specific limitations were set for the reference and literature search. The primary inclusion criteria encompassed scientific studies that utilized radiomics techniques, artificial intelligence (AI), or deep learning to classify osteoporosis in a diverse range of CT studies and compared their results to those of conventional DXA studies when possible. Articles excluded from further analysis comprised case reports, editorial correspondence (such as letters, commentaries, and opinion pieces), and review articles. Publications focusing on non-imaging radiomics techniques or articles that did not employ AI technology to classify osteoporosis from CT images were also excluded from the analysis.

### 2.3. Data Extraction and Reporting

All selected research articles were retrieved and compiled into a spreadsheet using Microsoft Excel Version 16.78.3 (Microsoft Corporation, Washington, DC, USA). Information gathered from the individual research articles included:Research article details: complete authorship, date of journal or publication, and journal name;Main clinical use: classify osteoporosis (either normal vs. abnormal BMD or normal vs. osteopenia vs. osteoporosis);Research study details: type of study, patient or imaging modality, body parts scanned, and area of bone segmented for analysis (e.g., internal or external data sets);Machine learning techniques used: radiomics, artificial or convolutional neural networks, etc.;Performance compared with DEXA: for example, the sensitivity, specificity, accuracy, correlation coefficients, and AUCs were obtained when possible.

## 3. Results

### 3.1. Search Results

The preliminary search of the main electronic medical databases (Figure 1) identified a total of 87 relevant articles, which were screened using the aforementioned criteria. This screening led to the initial exclusion of eight publications, and the remaining 79 articles underwent further full-text analysis to determine inclusion. Upon detailed analysis of the text, a further 48 publications were removed, as they either did not focus on the classification of osteoporosis or did not utilize AI methods. An additional eight articles were included after manually reviewing the bibliography of the selected articles. Overall, this culminated in a total of 39 articles (Figure 1) for in-depth analysis. The key findings were compiled and summarized in this review (Table 1). Most studies lacked detailed data to create 2 × 2 contingency tables, and hence a formal meta-analysis could not be performed.

Our search identified that of the 39 studies, 22 (56.4%) focused on unenhanced CT, whereas the remaining 17 (43.5%) investigated a combination of enhanced and unenhanced CT studies. Regarding the types of CT studies analyzed, 18 (46.1%) utilized CT abdomen and/or pelvis scans or related CTs encompassing the lumbar spine. Additionally, six (15.4%) studies were centered on CT scans of the thorax, including low-dose-screening CT. Two (5.1%) studies concentrated on CT coronary artery calcium scoring (CTCA), one of which also incorporated low-dose CT thorax in its analysis. A total of 10 (25.6%) studies were conducted on spine CTs, including one on the cervical spine and two on the thoracolumbar spine; the remainder were primarily on the lumbar/lumbosacral spine. The remaining 3/39 (7.7%) studies pertained to CTs of extremities or areas that were not specified. The majority of studies (21/39, 53.8%) analyzed one or more lumbar vertebrae to classify osteoporosis, while 6/39 (15.4%) focused on the thoracic vertebrae, 3/39 (7.7%) focused on the thoracolumbar vertebrae, and 1/39 (2.6%) focused on the cervical vertebrae. The remaining 4/39 studies (10.2%) assessed other non-axial bones, such as the knees, ribs, and wrists, while 2/39 (5.1%) analyzed soft tissues/muscles to classify osteoporosis.

The overall accuracy, sensitivity, and specificity of AI in classifying osteoporosis ranged from 61.8% to 99.4%, 41.0% to 100.0%, and 31.0% to 100.0% respectively, with AUCs ranging from 0.582 to 0.994. Of note, studies with two-label classification (normal versus abnormal BMD) achieved relatively higher performance in general compared with studies with three-label classification (normal vs. osteopenia vs. osteoporosis). Subdividing the analyzed regions showed that features from the lumbar vertebra appeared to achieve the highest AUC of 0.994. This is likely because binary classification is inherently simpler for AI models to handle. The model only needs to decide between two classes, which can result in a more straightforward decision boundary [106]. Furthermore, it is easier for AI models to identify and learn discriminative features when distinguishing between normal and abnormal BMD. In three-label classification, the model must discern finer distinctions, which can be more challenging and may require a larger, more complex model [107]. It is crucial to stress that the choice between binary (two-label) and ternary (three-label) classification should be guided by the specific clinical or research objectives. While binary classification may enhance accuracy, ternary classification can provide more intricate clinical insights and guide nuanced treatment decisions. Finally, it is worth noting that studies that have focused on the cervical and thoracic vertebrae have demonstrated only a moderate degree of correlation (*r =* 0.270 to 0.670) in contrast to studies encompassing the lumber vertebra (*r* = 0.582–0.911). The rest of the results for various subgroups are summarized in Table 2.

### 3.2. Artificial Intelligence

Artificial intelligence, often abbreviated as AI, involves harnessing the computational abilities of machines to carry out tasks resembling human activities [108]. This encompasses utilizing specific inputs to create outcomes that hold potential additional value [109]. Recent advancements in medical imaging alongside the accumulation of significant quantities of digital images and reports [110,111] have ignited increased worldwide interest in implementing AI in the medical imaging domain [112]. Initially conceived to aid radiologists in identifying and evaluating potential irregularities, both AI and computer-aided diagnostic (CAD) systems focus on amplifying efficiency, enhancing detection rates, and minimizing errors [113,114]. As a result, dedicated initiatives are striving to enhance AI’s diagnostic capabilities and optimize its efficiency for seamless integration into clinical practice. The emergence of convolutional neural networks, inspired by the mechanisms of the human brain, has introduced a range of computational learning models primarily centered around machine learning (ML) and deep learning (DL) algorithms [115]. These models have played a pivotal role in propelling the widespread adoption of AI in radiology.

### 3.3. Machine Learning, Deep Learning, and Radiomics

Machine learning (ML) is a branch of artificial intelligence (AI) that involves training models to make predictions on the basis of existing datasets. These models use their learned knowledge to perform tasks on new, unfamiliar data. To apply ML, data inputs are collected and labeled by experts or extracted using computational methods. Supervised ML models learn from data labeled by human experts to make predictions or classify information, while unsupervised models learn from unlabelled data to uncover patterns and relationships within datasets. Unsupervised models can be used to represent datasets more efficiently and understand their inherent structures. This representation can be a preliminary step before training a supervised model, potentially enhancing its performance.

Deep learning (DL), a subset of ML (Figure 2), mimics the structure of neural networks in the brain. It employs artificial neural networks with multiple hidden layers to solve complex problems. These hidden layers enable the system to continuously learn and improve its performance by incorporating new knowledge. Unlike traditional ML, which requires manual extraction of features from input images, DL methods learn features directly from input images using multilayer neural networks like convolutional neural networks (CNNs). This approach allows DL systems to not only map image features to outputs but also learn the features themselves. Examples of DL outputs include image categories (classification), object locations (detection), and pixel labels (segmentation). Deep learning has given rise to the field of radiomics, which entails extracting a multitude of quantitative features from medical images, including CT scans, to unveil hidden patterns through computational analysis. In the context of bone health, radiomics employing deep learning techniques has the potential to identify imaging features related to important pathological and histological characteristics of bone trabeculae. These features can surpass human diagnostic capabilities and potentially outperform conventional imaging methods. The primary machine learning methods in use are radiomics-based feature analysis (Figure 3), which involves manually designed feature extraction integrated into deep learning training datasets [116], and convolutional neural networks (CNN), which automatically extract valuable image features to classify data directly from input images [117]. CNNs enable the identification of diagnostic patterns and features that exceed human capacity and have applications in osteoporosis diagnosis and classification [118,119].

### 3.4. General Workflow of BMD Classification in CT

The main methods for extracting valuable bone mineral density (BMD) information from CT images involve the conversion of Hounsfield Units (HUs) measured within the bones [97,98,99]. The HU is a relative quantitative measure of radiodensity used in CT scans to quantify the density of specific tissues or substances within the body. It provides information about the attenuation of X-rays as they pass through the tissue, with calibrated values defined by the densities of air and water as reference points.

After deriving HU values from bones, BMD values can be computed using two main approaches:Phantom-Based Calibration: This method involves placing phantoms containing known densities (such as dipotassium phosphate or calcium hydroxyapatite density rods) beneath the subject during scan acquisition [62,101]. These phantoms are used to calibrate measured HU values to BMD through linear equations. One challenge is that phantom placement is not routinely performed in clinical CT scans [102]. However, this challenge can be overcome by scanning the density phantom asynchronously using the same scanner and scan protocol but without the patient present.Phantomless Internal Calibration: In this approach, the HU peak values of internal reference regions, such as skeletal muscle and adipose tissue, are used. The reference BMD density values for these internal references are determined using phantom-calibrated scans from a cohort of patients [103]. These values are then extrapolated to create a standard calibration curve for converting the trabecular HU to BMD [104,105] through scan-specific equations.

Another reported method for determining BMD from CT images involves applying HU thresholds for osteoporosis screening [48,61,106,107,108]. However, this method lacks reliability due to HU sensitivity to X-ray energy, beam hardening artifacts, positioning, and hardware-related variations, including different scan models and protocols [109,110,111].

To enhance BMD prediction and classification, artificial intelligence (AI) and deep learning methods are increasingly employed. These methods extract imaging data, including CT attenuation (HU) of bones and texture features, to create models [112,113]. The standard process for developing an AI model involves image acquisition and data selection, segmentation, the extraction of image features within specified regions of interest (ROIs), exploratory analysis with feature selection, and building the model [114,115]. The models are validated using test sets, ideally comprising both internal and external data, to assess their performance and generalizability [116]. The two primary machine learning methods used are:Feature-Based Imaging Feature Analysis: This approach involves manually extracting various features and incorporating them into a training set for AI-based imaging classification [117].Deep Learning-Based Analysis (e.g., CNNs): CNNs employ deep learning to automatically extract valuable imaging features by learning patterns directly from input images [118]. This enables the detection and processing of distinct diagnostic patterns and imaging features that go beyond what a human reader can accomplish [120], potentially improving BMD classification.

**Table 1 bioengineering-10-01364-t001:** Key characteristics of the selected articles.

Authors	Artificial Intelligence Method	Publication Year	Main Objectives	Title of Journal	Main Type of CT	Areas Sampled	Performance
Yasaka K. et al. [66]	CNN	2020	Predict osteoporosis	European Radiology	Unenhanced CT of abdomen	L1 vertebra	*r* = 0.852 (*p* < 0.001), AUC = 0.965 (internal validation) 0.840 (*p* < 0.001), AUC = 0.970 (external validation)
Kang J.W. et al. [61]	ResNet-101v2, CNN	2023	Classify osteoporosis	Frontiers in Physiology	Unenhanced CT of abdomen	L1 vertebra	*r* = 0.900 F1 score = 0.875
Uemura K. et al. [120]	Computer-aided system	2023	Classify osteoporosis	Archives of Osteoporosis	Unenhanced CT of abdomen	Axial slice of the L1 vertebra (L1-vBMD) Axial slices of L1–L4 (CT-vBMD) Coronal L1–L4 (CT-aBMD)	*r* = 0.364, AUC = 0.582 (L1-vBMD); *r* = 0.456, AUC = 0.657 (CT-vBMD); *r* = 0.911, AUC = 0.941 (CT-aBMD)
**Savage R.H. et al.** [121]	Wavelet features, AdaBoost, and local geometry constraints	2020	Classify osteoporosis	Journal of Thoracic Imaging	Unenhanced CT of thorax	Thoracic vertebrae	Moderate correlation, *r* = 0.55 (*p* < 0.001) Significant difference between normal control patients and osteoporotic group (*p* = 0.045)
**Pickhardt P.J. et al.** [101]	CNN (U-Net, TernausNet)	2022	Classify osteoporosis	Radiology	CT of abdomen	L1 bone (one to seven slices)	AUC = 0.860–0.930 Sensitivity: 85.4%–94.0%; Specificity: 94.6%–98.3%. Accuracy: 89.0%–94.0%
Fang Y. et al. [95]	CNN (DenseNet-121), U-Net	2021	Classify osteoporosis	European Radiology	CT of abdomen and CT of spine	L1–L4 vertebrae	*r* > 0.980 (*p* < 0.001) Cohen’s kappa = 0.868–0.888)
Pan Y. et al. [122]	U-Net	2020	Classify osteoporosis	European Radiology	Low-dose CT of thorax	T1–L2 vertebrae	*r* = 0.964–0.968 Mean errors: 2.2–4.0 mg/cm AUC = 0.927 (osteoporosis), 0.942 (low BMD)
Tang C. et al. [123]	CNN (MS-Net, BMDC-Net)	2021	Classify osteoporosis	Osteoporosis International	CT of abdomen or lumbar spine	L1 vertebra	Accuracy: 76.7% AUC = 0.917
Dzierżak, R. et al. [93]	Deep CNN (VGG16, VGG19, MobileNetV2, Xception, ResNet50, and InceptionResNetV2	2022	Classify osteoporosis	Sensors	CT of lumbosacral spine	L1 vertebra	AUC = 0.883–0.973 Accuracy: 84.0%–95.0% Sensitivity: 78.0%–96.0% Specificity: 86.0%–98.0%
Breit H.C. et al. [124]	CNN	2023	Classify osteoporosis	European Journal of Radiology	Non-contrast CT of thorax	Thoracic vertebrae	*r* = 0.51, *p* < 0.001 (hip BMD); *r* = 0.34, *p* = 0.01 (lumbar spine BMD) Accuracy: 75.0%, Sensitivity: 93.0%, Specificity: 61.0%; Significantly better than clinical reports
**Summers R.M. et al.** [125]	Computer-aided Software (QCT Pro software, versions 3.2, 4 or 4.1)	2011	Classify osteoporosis	Journal of Computer Assisted Tomography	CT, colonoscopy	L1–L2 vertebrae	*r* = 0.980 (*p* < 0.0001) 95% limits of agreement were (−9.79, 8.46) mg/cc
Valentinitsch, A. et al. [126]	RF classifier	2019	Classify osteoporosis	Osteoporosis International	CT of thoracolumbar spine	Thoracolumar vertebrae	AUC = 0.71–0.88
Sebro R. et al. [103]	Naïve Bayes; RF; SVM; XGBoost	2023	Classify osteoporosis	Journal of Neuroradiology	CT of cervical spine	C1–T1 vertebrae	AUC = 0.622–0.843 Accuracy: 74.6%–99.4% Sensitivity: 85.0%–100% Specificity: 56.7%–98.5%
Sebro R. et al. [127]	(LASSO), Elastic Net, Ridge regression, and SVM with RBF	2022	Prediction of osteoporosis	European Journal of Radiology	CT of thorax	Ribs, thoracic vertebrae, sternum, and clavicle	*r* > 0.4, *p* < 0.001 AUC = 0.702–0.757
Liu et al. [128]	LR, SVM with RBF, ANN, RF, eXtreme Gradient Boosting and Stacking	2022	Classify osteoporosis	BMC Bioinformatics	CT images covering lumbar vertebral bodies	L1–L4 vertebrae	AUC = 0.818–0.962 Accuracy: 86.6%–96.0% Sensitivity: 71.6%–96.4% Specificity: 91.6%–96.0%
Pan J. et al. [100]	ResNet-101 residual DCNN classification model	2023	Classify osteoporosis	Research Square	CT of thorax	L1–L2 vertebrae, L1 vertebra	AUC = 0.940–0.990 Accuracy: 86.6%–94.7% Sensitivity: 63.8%–97.6% Specificity: 83.8%–93.0%
Lim H.K. et al. [129]	Aquarius iNtuition v4.4.121, TeraRecon, Medip, RF	2021	Classify osteoporosis	PloS ONE	Unenhanced CT of abdomen and pelvis	Left femur	AUC = 0.959–0.960 Accuracy: 92.7%–92.9%; Sensitivity: 80.0%–86.6%; Specificity: 94.5%–95.8%;
Zhang K. et al. [130]	CNN	2023	Classify osteoporosis	Computational Intelligence and Neuroscience	CT images covering lumbar vertebral bodies	L1–L2 vertebra	AUC = 0.965–0.985 Accuracy: 93.3%–97.1% Sensitivity: 83.6%–96.4% Specificity: 92.2%–97.6%
Nam K.H. et al. [131]	MR, LR. Tensor flow and Python	2019	Classify osteoporosis	Journal of Korean Neurosurgical Society	CT of lumbar spine	L1–L3 vertebra	AUC = 0.900 Accuracy: 92.5% F1 score: 0.954;
Xu Y. et al. [132]	SVM and kNN	2013	Classify osteoporosis	Microscopy Research Technique	Micro-CT	-	F1 score: 0.900–0.958 Precision: 91.3%–95.3%
**Löffler, M.T. et al.** [98]	CNN	2021	Classify osteoporosis	European Radiology	CT of lumbar spine	L1–L4	AUC = 0.860–0.885 Sensitivity: 41.0%–86.0% Specificity: 78.0%–98.0% (superior to DXA for predicting osteoporosis in patients with vertebral fractures
Krishnaraj A. et al. [133]	Machine learning-based regression	2019	Classify osteoporosis	Journal of American College of Radiology	CT of abdomen and pelvis	L1–L4 vertebrae	Accuracy: 82.0%; Sensitivity: 84.4%; Specificity: 72.7%
Chen Y.C. et al. [134]	CNN (ResNet50), SVM	2023	Classify osteoporosis	European Radiology	Low-dose CT of thorax	Thoracic vertebrae	AUC = 0.960–0.980 Accuracy: 85.0%–95.0% Sensitivity: 85.0%–94.0% Specificity: 85.0%–92.0%
Tariq A. et al. [135]	CNN (Densenet121)	2023	Classify osteoporosis	Medical Physics	Contrasted/non-contrasted CT of abdomen and pelvis	L3 vertebrae	AUC = 0.830 (axial), 0.830 (coronal), 0.860 (imaging + demographic factors)
Elmahdy, M. et al. [94]	SVM with RBF	2023	Classify osteoporosis	Studies in Health Technology and Informatics	CT of knee	Distal femur, proximal tibia and fibula, and patella	AUC = 0.937 Sensitivity: 83.3% Specificity: 100.0%
Sollmann N. et al. [136]	CNN (DenseNet)	2022	Classify osteoporosis	Journal of Bone Mineral Research	CT of abdomen and pelvis	T6 to L5 vertebrae	AUC = 0.815–0.862
Yang J. et al. [69]	CNN	2022	Classify osteoporosis	Osteoporosis International	CT of thorax	Thoracic vertebrae	AUC = 0.831–0.972 Sensitivity: 73.8%–95.6% Specificity: 73.6%–88.0%
Sebro R. et al. [137]	SVM with RBF	2022	Classify osteoporosis	Diagnostics	CT of wrist/forearm	Forearm, carpal, and metacarpal bones	AUC = 0.818 (radius) Sensitivity: 69.2% Specificity: 77.1% *r* = 0.74–0.85
Yoshida K. et al. [138]	CNN (ResNet50)	2023	Classify osteoporosis	Journal Computer Assisted Tomography	Non-contrasted CT images covering lumbar vertebral bodies	L1–L4 vertebrae	AUC = 0.921–0.969 *r* = 0.81 Accuracy: 73.0%–94.0% Sensitivity: 73.0%–100% Specificity: 73.0%–94.0%
Dai H. et al. [139]	LASSO regression model	2023	Classify osteoporosis	Acta Radiological	CT of abdomen	Lumbar vertebrae	*r* = 0.932
**Huang C.B. et al.** [96]	LASSO, GNB, RF, LR, SVM, GBM, XGBoost	2022	Classify osteoporosis	BMC Geriatrics	CT of abdomen	Psoas at L3 level	AUC = 0.860 Accuracy: 81.0% Sensitivity: 70.0%, Specificity 92.0%
Naghavi M. et al. [140]	CNN (Unet)	2023	Classify osteoporosis	Journal of the American College of Radiology	CT, coronary artery calcium scoring	Thoracic vertebrae	*r* = 0.84 AutoBMD averaged 15 s per report vs. 5.5 min for manual measurements (*p* < 0.0001).
Naghavi M. et al. [99]	CNN (Unet)	2023	Classify osteoporosis	European Journal of Radiology Open	Low-dose CT of thorax, CT, coronary artery calcium scoring	Thoracic vertebrae	R2 = 0.95 (*p* < 0.0001) Similar results in both modalities
Küçükçiloğlu Y. et al. [97]	CNN (InceptionV, EfficientNetV2S, ResNet50	2023	Classify osteoporosis	Diagnostic Interventional Radiology	CT of lumbar spine MRI of lumbar spine	Lumbar vertebrae	AUC = 0.942–0.988(CT) Accuracy: 98.8% (CT) Sensitivity: 98.5% (CT) Specificity: 99.2% (CT) AUC = 0.980 (CT + MRI) Accuracy: 96.8% (CT + MRI) Sensitivity: 96.7% (CT + MRI) Specificity: 96.8% (CT + MRI)
Wang J. et al. [104]	PyRadiomics, LASSO	2023	Classify osteoporosis	BMC Musculoskeletal Disorder	CT of lumbar spine	L1 vertebra	AUC = 0.902–0.988 Accuracy: 86.0%–94.0% Sensitivity: 85.7%–87.5% Specificity: 80.0%–97.2%
**Jiang, Y.W. et al.** [105]	mRMR, LASSO	2022	Detect osteoporosis	European Radiology	CT of lumbar spine	L1 vertebra	AUC = 0.762–0.969 Accuracy: 75.9%–87.1% Sensitivity: 59.5%–73.0% Specificity: 83.5%–93.7%
Xue Z. et al. [141]	PyRadiomics, SVM, RF, KNN	2022	Detect osteoporosis	BMC Musculoskeletal Disorder	CT of lumbar spine	L1–L4 vertebrae	AUC = 0.994 (normal vs. osteoporosis) AUC = 0.866 (osteopenia vs. osteoporosis) AUC = 0.940 (normal vs. osteopenia)
Qiu H. et al. [142]	mRMR, LASSO	2022	Detect osteoporosis	Frontiers in Endocrinology	CT covering lumbar vertebra bodies	Paravertebral muscles at the level of the L1 vertebra	AUC = 0.900 (radiomics); 0.950 (radiomics + clinical features) Accuracy: 81.4%–88.1% Sensitivity: 85.7%–88.9% Specificity: 77.4%–87.5%
Mookiah M.R.K. et al. [143]	SVM	2018	Classify osteoporosis	Osteoporosis International	CT images covering thoracolumbar spine	Thoracolumbar spine	Accuracy: 83.0% Sensitivity: 93.3% Specificity: 79.3% *r* = 0.91–0.96

Area under the curve (AUC), random forest (RF), support vector machine (SVM), artificial neural network (ANN), convolutional neural network (CNN), radial basis function (RBF), logistic regression (LR), multiple regression (MR), k-nearest neighbor (kNN), least absolute shrinkage and selection operator (LASSO), Gaussian naïve Bayes (GNB), gradient boosting machine (GBM), minimum redundancy–maximum relevance (mRMR).

**Table 2 bioengineering-10-01364-t002:** Summary of Results.

Areas Sampled	No. of Studies	Area under the Curve (AUC)	Accuracy	Sensitivity	Specificity	*r*
Cervical vertebrae	1	0.622–0.843	74.6%–99.4%	85.0%–100%	56.7%–98.5%	0.270–0.670
Thoracic vertebrae	6	0.831–0.980	85.0%–95.0%	73.8%–95.6%	73.6%–92.0%	0.34 0–0.510
Thoracolumbar vertebrae	4	0.710–0.952	83.0%	93.0%	79.3%	0.910–0.968
Lumbar vertebrae	21	0.582–0.994	73.0%–98.8%	41.0%–100%	73.0%–99.2%	0.582–0.911
Other regions	7	0.630–0.960	61.8%–92.9%	61.8%–95.0%	31.0%–100%	0.400–0.600
Overall	39	0.582–0.994	61.8%–99.4%	41.0%–100%	31.0%–100%	0.270–0.968

## 4. Discussion

### 4.1. Advantages and Efficacy

CT examinations performed for other indications present a unique opportunity for incidental osteoporosis screening with no additional cost, time penalty, or radiation exposure for patients [74,144,145]. Many studies have already shown that information from a single L1 vertebra body on CT correlates well not only with T-scores from DEXA [74] but also with the risk of future osteoporotic fractures [146,147,148]. The wealth of data obtained from this straightforward evaluation of trabecular attenuation values in CT scans can even compete with the predictive capabilities offered by the more cumbersome FRAX (fracture risk assessment tool) method [149,150]. The FRAX method is a widely used clinical tool that estimates the 10-year probability of fractures on the basis of various clinical risk factors and bone mineral density measurements. Despite its prevalence, the simplicity and accuracy of trabecular attenuation values from CT scans present a compelling alternative for evaluating bone health.

The utilization of CT images for osteoporosis prediction and classification has been enabled by recent advancements in AI and deep learning techniques. Deep learning methods like CNNs enhance the accuracy of osteoporosis diagnosis by constructing multi-hidden-layer models and leveraging extensive training data sets to identify, extract, and learn valuable features. These methods provide unique insights into bone quality and microstructure that cannot be attained solely through DEXA scans or visual assessment, ultimately elevating the accuracy of osteoporosis prediction and classification. Yasaka et. al. [66] revealed that a BMD model using CNN on CT images was slightly superior (AUC = 0.965–0.970) to past estimations of BMD using CT attenuation (AUC = 0.829–0.953) alone (*p* = 0.013). Their BMD model also showed a higher positive correlation to DEXA-based BMD estimation (*r* = 0.852 vs. *r* = 0.425 for CT attenuation alone, *p* < 0.001).

A crucial facet of AI technology lies in its proficiency for automated segmentation [151,152,153], which not only minimizes processing time but also improves the precision of region of interest (ROI) placement. This is especially advantageous in atypical cases in which manual segmentation is both time-consuming and challenging, leading to an enhancement in sensitivity when classifying osteoporosis in such complex scenarios. The deep learning-based BMD tool developed by Pickhardt et al. [84] demonstrated a significantly higher success rate in accurately placing and sizing regions of interest (ROIs) compared with an older automated feature-based algorithm (99.3% vs. 89.4%, *p* < 0.001). This difference was particularly evident among patients with suboptimal positioning, deformities like scoliosis, metallic hardware, or other structural variations that can impact BMD measurements when using DEXA.

Radiomics is a widely applied technique in clinical oncology for tasks such as cancer detection and diagnosis, forecasting outcomes and prognosis, and predicting post-treatment response [154,155,156]. Recently, several studies have emerged in which textural characteristics extracted from X-rays, MRI scans, and DEXA scans have been applied to detect, diagnose, and classify osteoporosis and other metabolic bone disorders [141,142,143,144]. However, the diagnostic accuracy for osteoporosis on these modalities remains inadequate, with reported values for the area under the curve (AUC) in osteoporosis classification typically hovering around 0.8. These limitations may be attributed to the choice of the region of interest (ROI), which often lacks a comprehensive 3D reconstruction of the entire vertebral body. As a result, this omission potentially results in the exclusion of pertinent information, thereby impacting the accuracy of the predictive model.

CT has the advantage of 3D evaluation of bone mineralization and allows for volumetric analysis of the entire vertebral body rather than the assessment of just a small sample of the vertebral body trabeculae. Volumetric measurement of BMD is known to be more sensitive and precise than DEXA for detecting bone mineral loss, as it avoids the superimposition of cortical bone and other soft tissues [157]. The model created by Xue et al. [141] achieved a high AUC of 0.994 in differentiating normal and osteoporotic bones using radiomics features extracted from volumetric analysis of the lumbar spine on CT images.

Measuring CT attenuation or the Hounsfield Units (HUs) of vertebral bodies has been extensively explored as a method for osteoporosis detection, given that it is an easily accessible tool on most PACSs [158] with the assistance of phantom and phantomless calibration. However, this approach has inherent limitations. Firstly, the calibration equation is susceptible to variations in scan protocols, phantom positioning, and, in the case of phantomless calibration, the patient cohort used [56,159]. As such, it cannot be applied retrospectively, and it may not be reproducible in other cohorts or institutions. By contrast, machine learning models incorporate the analysis of imaging features beyond CT attenuation alone and may even surpass conventional HU measurements alone. Jiang et al. [88] demonstrated the effectiveness of a CT-based radiomics signature generated through 3D feature extraction from the lumbar spine. This approach outperformed HU measurements alone, achieving an AUC of 0.960 (*p* < 0.05) and offering a higher overall net benefit, as determined by decision curve analysis.

AI facilitates the seamless integration of radiomics with clinical information to create a robust model for classifying BMD on the basis of CT images. Clinical data and demographic traits [160,161] have been employed to identify individuals with osteoporosis or osteoporotic fractures, playing a pivotal role in the development of various tools for osteoporosis assessment [162]. The inclusion of clinical characteristics (such as age, sex, and risk factors as per the National Osteoporosis Foundation Guidelines [145]) in BMD assessment using DEXA has previously shown promising results in predicting osteoporosis, as demonstrated by Wang et al. [104]. Their radiomics clinical model achieved an AUC of 0.988, compared with an AUC of 0.902 with radiomics alone (although this difference was not statistically significant, *p* = 0.643). Notably, a radiomics model that incorporates both clinical and CT imaging data has the potential to match or even surpass the performance of DEXA alone, which primarily focuses on bone density analysis. A study by Liu et al. [128] demonstrated that their logistic regression model including clinical and CT imaging data achieved superior performance with an AUC of 0.962 compared with using either clinical data alone (AUC = 0.819–0.828) or CT image features alone (AUC = 0.876–0.953). These findings emphasize the promise of AI-driven models that leverage both radiomics and clinical data for improved osteoporosis classification.

The primary objective of opportunistic osteoporosis screening via CT images is to leverage CT scans conducted for unrelated purposes to diagnose and classify osteoporosis. To date, no studies have compared the effectiveness of AI-assisted screening using CT scans for other indications against those specifically conducted for osteoporosis diagnostics, such as quantitative CT. However, we hypothesize that the latter methods may offer more accurate and detailed information, given their adherence to strict acquisition protocols [163,164]. Further studies are needed to evaluate this aspect, an important step before determining the viability of opportunistic osteoporosis screening using CT scans from diverse medical contexts. Later, this paper discusses various technical considerations for employing CT images from other indications in osteoporosis diagnosis and classification, providing valuable insights for future studies and model development.

#### 4.1.1. Technical Considerations: Labeling and Segmentation

An important aspect of automated segmentation lies in the precise labeling of bones and vertebral levels [163]. In the majority of CT scans of the chest or abdomen, the entire spine is not fully captured, and the vertebra of interest may be at the edge of the field of view, resulting in partial inclusion. Consequently, the model’s capability should not rely on the specific number or position of the vertebrae within the scan. Instead, it should consider alternative features, such as rib count [150], for identification and segmentation. Accurate labeling of vertebral levels is important as it has been demonstrated that BMD can decrease from the thoracic to the lumbar spine [164,165]. Furthermore, the task of correct vertebral labeling can be further complicated by the presence of transitional vertebrae [32,153]. Developing a robust AI model that includes such complex cases within its training set may offer a potential solution to this challenge.

#### 4.1.2. Technical Considerations: Contrast versus Non-Contrast

The majority of clinical CT scans employ contrast agents to examine structures such as blood vessels, soft tissues, and other internal body organs for various medical conditions. However, the use of contrast alters the way tissues absorb X-rays, resulting in an approximate 8–10% increase in the CT attenuation of trabecular bone [154] compared with unenhanced scans [155,156]. Notably, the impact of contrast on bone trabeculation varies depending on the region of interest sampled, primarily due to differences in blood supply and vasculature. Bauer et al. [166] demonstrated that contrast enhancement increased synchronous phantom-derived BMD values by approximately 31% in the lumbar spine and 2% in the proximal femur compared with unenhanced CT scans.

Several studies have explored the use of internal calibration to address differences between contrasted and non-contrast studies, but these efforts have encountered limited success [158,159]. Key challenges have been identified, including the heterogeneity in marrow enhancement caused by multiple factors, such as the time elapsed between contrast administration and CT scan acquisition, heterogeneity in marrow enhancement due to differences in bone mineral density, and other contributing factors [160]. With the emergence of AI and machine learning, models can now be trained to incorporate both enhanced and unenhanced CT scans, enabling them to adapt their prediction algorithms. This adaptation may involve adjusting CT attenuation thresholds for patients with contrast-enhanced CT studies [112] or even identifying the contrast phase to correct for variations [73,161]. This approach could overcome the need for a one-size-fits-all internal calibration, making it applicable to a wider range of datasets.

#### 4.1.3. Technical Considerations: Areas Sampled

The location and types of bones selected for sampling are recognized to give varying BMD results, a phenomenon also observed in DEXA studies [167]. In our study, we found that investigations focusing on the thoracolumbar or lumbar spine (Table 2) generally exhibited stronger positive correlations, reaching up to *r* = 0.968 and *r* = 0.911, respectively, in comparison with examinations of other anatomical regions (such as cervical vertebrae, with *r* = 0.670; thoracic vertebrae, with *r* = 0.510; and other body parts, with *r* = 0.600) in relation to BMD results acquired via DEXA. This trend is likely due to the proximity of the thoracolumbar and lumbar vertebrae to the areas typically sampled by DEXA (usually the L1–L4 vertebrae) for BMD classification and osteoporosis assessment. Even though variations in BMD exist across different bones, meaningful comparisons remain feasible between various skeletal sites due to the strong correlations established between them [163,164,165,166].

Evaluating non-spinal skeletal sites on CT for osteoporosis represents a valuable clinical approach to assessing bone health beyond traditional spine assessment. Analyzing peripheral sites, such as the hip or wrist, could provide clinicians with a comprehensive understanding of a patient’s overall skeletal integrity [168,169]. Furthermore, non-spinal skeletal sites may be subjected to fewer degenerative changes [170], enhancing their predictive value for BMD and future fracture risk. The sites evaluated in the literature include the hip, distal femur [129], tibial, fibula [94], distal radius, hand bones [137], ribs, sternum, and clavicle [127]. This broader assessment can be particularly useful in cases in which spine imaging is not performed or when it will not provide a complete picture of the patient’s bone health, e.g., vertebral degeneration, fractures, or instrumentation precludes accurate assessment. Additionally, scanning these sites exposes essential organs to less radiation.

Detailed information about localized bone health at these peripheral sites can prove important in guiding treatment decisions. For instance, Gruenewald et al. [34] demonstrated the opportunistic evaluation of BMD during planning CT scans for distal radius fracture fixation. This approach helped predict outcomes and the need for surgical bone substitutes during fixation, especially for patients with a mean volumetric BMD lower than 79.6 mg/cm^3^ who required surgical bone substitutes during fixation, as well as patients with a mean volumetric BMD lower than 71.1 mg/cm^3^ who developed bone non-union (AUC = 0.710–0.910). This promising approach may extend to other common fracture sites such as the femoral neck, helping predict the need for surgical treatment and guiding interventions to enhance patient outcomes.

Interestingly, two studies by Huang CB et al. [79] and Qiu H. et al. [127] adopted a unique approach by sampling muscles rather than the bony skeleton to predict and classify osteoporosis. This strategy capitalizes on the established link between sarcopenia and osteoporosis [167,168,169]. Prior studies have shown that individuals with both sarcopenia and low BMD have an increased risk of insufficiency fractures [129,170]. The radiomics model developed by Huang CB et al., which employed gradient boosting methods (GBMs), achieved an AUC of up to 0.860 and an accuracy of 81.0% on validation sets [79]. This method of classifying osteoporosis is especially useful in cases in which direct sampling of the vertebral body may not provide an accurate representation of bone density, for example, in cases of severe spinal spondylosis [171].

The accuracy of a machine learning model utilizing CT attenuation values from multiple bones in conjunction with clinical and demographic variables exceeded that of models relying on a single bone. Uemura K. et al. [120] demonstrated that their model, when limited to sampling just the L1 vertebral region, achieved an AUC of 0.582, significantly lower than when they expanded their sampling to include the L1–L4 vertebrae (AUC = 0.941). Similarly, Sebro R et al. [137] showed that using data from multiple bones in the wrist yielded superior accuracy in contrast to relying on CT attenuation values from a single bone. These observations suggest that future comparative studies should be performed across various skeletal areas to determine the optimal region(s) and the extent of sampling for enhanced accuracy.

### 4.2. Other Potential Applications: Incorporating Molecular and Genetic Biomarkers

Recent advances in molecular diagnostics have shown remarkable potential in the realm of bone mineral density (BMD) and osteoporosis diagnosis [172,173]. In particular, testing for bone turnover markers (BTMs) detects peptides produced during bone matrix formation and degradation [174,175]. These are substances found in the blood and urine, providing information about the rate at which bone tissue is broken down (resorption) and formed (formation). Notably, BTMs such as PINP (N-terminal propeptide of type I procollagen) and CTX (C-telopeptide of type I collagen) [176] can offer early detection of bone loss before it becomes severe, which is particularly valuable in identifying individuals who may benefit from preventive measures and early intervention. This is possible because BTMs reflect the dynamic process of bone remodeling, thereby providing real-time information about bone health before changes can be detected by conventional DEXA scans [177,178]. However, these molecular tests and genetic analyses can be costly and may require specialized laboratories [179,180]. Furthermore, they are subject to significant pre-analytical and analytical variability, with a lack of standardization for BTM assays [181].

AI has the potential to assist in the molecular diagnosis and genetic analysis of osteoporosis when applied to CT scans. While relatively new in the field of bone health, the use of AI to analyze imaging features to determine genomic signatures and advanced biomarkers has been extensively studied in the realm of oncology, known as radiogenomics [182,183]. For instance, Ren et al. [184] and Fan et al. [185] were able to predict EGFR mutation status using radiomics analysis on vertebral metastases from lung cancer, eliminating the need for actual testing. Xu R. et al. [186] were able to develop a radiomics model to predict molecular biomarkers such as estrogen receptors (ERs), progesterone receptors (PRs), and human epidermal growth factor receptor 2 (HER2) status using ultrasound images of the breast, which otherwise can only be obtained by biopsy or surgery. Similarly, AI models could be trained to predict molecular markers (such as BTMs) and genetic markers from CT images without the need for formal testing. The combination of AI with CT diagnostic methods and molecular-level assessments has the potential to enhance our understanding of osteoporosis, improve early detection, enable personalized treatment strategies, and ultimately reduce the burden of this disease on individuals and healthcare systems.

### 4.3. Challenges in Implementation

Despite achieving numerous promising outcomes, the integration of various AI methods for CT screening of osteoporosis into clinical practice faces several challenges that need addressing. Firstly, AI development requires a substantial volume of medical imaging data, raising concerns about data ownership, usage for research, informed consent, and patient confidentiality, often contingent on local legal frameworks [187]. In addition, the training and validation of medical image algorithms involve a time-intensive and costly process of labeling numerous parameters, typically performed by radiologists [188]. Relying on smaller datasets from a single institution can yield unreliable results when applying the AI model to a different population, posing challenges related to generalizability and reproducibility [189,190]. To mitigate these issues, external validation and the use of large multicentre databases as benchmarks are necessary to enhance AI model performance before clinical implementation [191,192,193]. However, coordinating such extensive projects can be challenging.

Additionally, the seamless integration of AI models into the clinical workflow and interface with existing radiology information systems (RISs) and picture archiving and communication systems (PACS) is crucial [194]. This integration can be problematic, especially due to variations in IT environments across different institutions [195] and the absence of standardized protocols for data sharing between digital systems [196].

## 5. Conclusions

This systematic review highlights the growing body of evidence that underscores the promise of harnessing artificial intelligence in tandem with CT scans for osteoporosis screening and classification. The synergy between advanced imaging technologies and AI algorithms presents an opportunity to revolutionize osteoporosis diagnosis and risk assessment. Our study highlights various key considerations for the use of CT imaging as an opportunistic screening tool for osteoporosis, facilitated by AI assistance. These insights may be helpful for forthcoming research, spanning model development through to clinical integration.

Most of the research conducted in this field has consisted of preliminary investigations, retrospective analyses, or studies conducted at single centers, often with small sample sizes. Consequently, the models developed in these studies have limited applicability, and when applied to external datasets, they often yield variable results due to significant heterogeneity. This variability hampers the ability to reproduce results consistently and hinders the development of AI models suitable for clinical implementation. To address this issue, additional research, particularly randomized controlled trials or large-scale multi-center studies, is essential to validate these applications and pave the way for their seamless integration into standard clinical practice.

This study also discussed the significance of technical considerations when analyzing CT images for opportunistic osteoporosis diagnosis in non-osteoporosis contexts, offering valuable insights for future model development. Furthermore, the incorporation of clinical characteristics and radiomics features in AI-based osteoporosis diagnosis may outperform conventional methods across diverse clinical settings. While this shows promise, further research should address clinical and implementation aspects before clinical translation is feasible.

Finally, our article offers a comprehensive review of the available evidence regarding the use of CT images for classifying osteoporosis. However, it is important to acknowledge the inherent limitations of a scoping review, such as the absence of detailed data extraction, the omission of statistical analysis, and the lack of a formal quality assessment for the included studies, which may have introduced subjectivity and potential bias into our findings. Nevertheless, our review article should serve as a valuable resource for future research projects in this field.

## Figures and Tables

**Figure 1 bioengineering-10-01364-f001:**
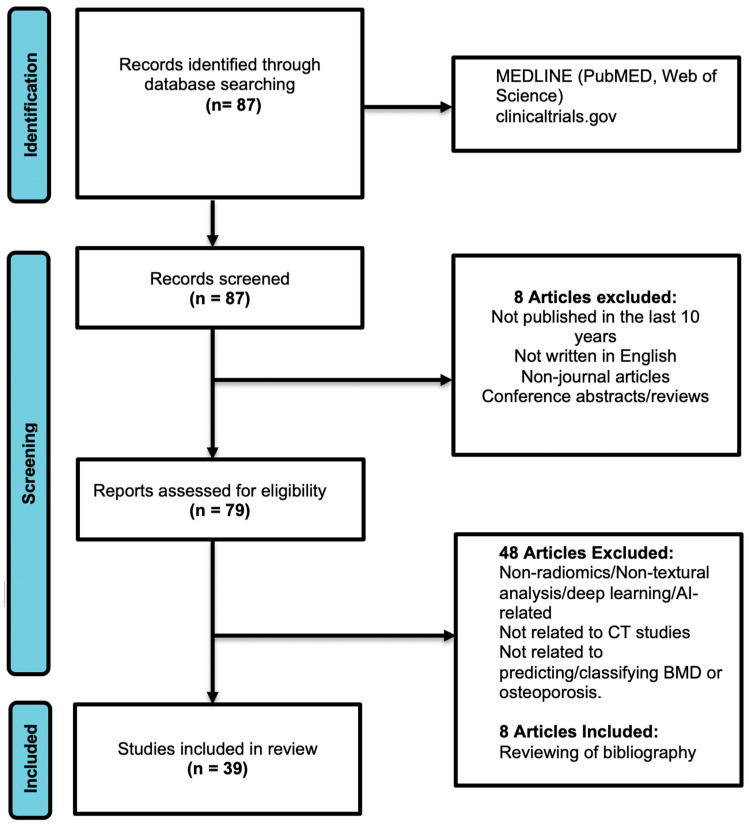
PRISMA flowchart for the literature search (this is adapted from the PRISMA group, 2020), which describes the selection of relevant articles.

**Figure 2 bioengineering-10-01364-f002:**
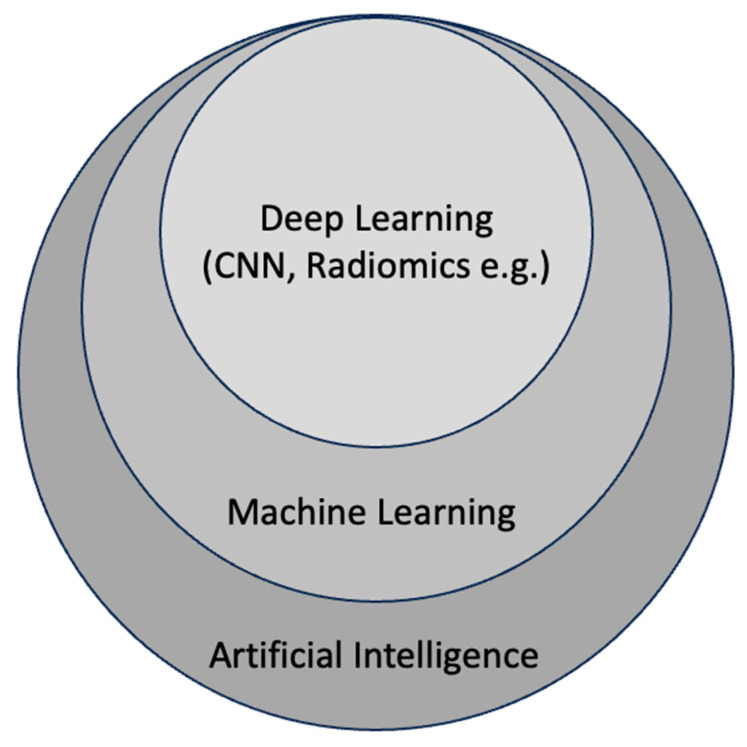
Diagram illustrating the hierarchical structure of artificial intelligence. Machine learning, a subset of artificial intelligence, is a discipline that imparts computers with the capacity to learn autonomously, bypassing the need for explicit programming. Within the realm of machine learning, deep learning represents a subset that enables the computation of neural networks with multiple layers. CNN is a subset of deep learning characterized by convolutional layers.

**Figure 3 bioengineering-10-01364-f003:**
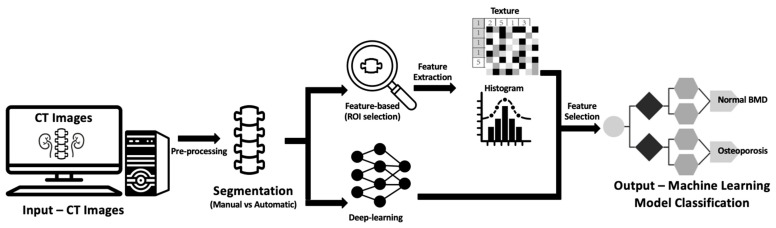
Diagram showing the general framework and main steps of radiomics, namely data selection (input), segmentation, feature extraction in the regions of interest (ROIs), exploratory analysis, and modeling.

## Data Availability

Not applicable.

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
