# Peer review of "Artificial Intelligence Applications for Osteoporosis Classification Using Computed Tomography"

_bioengineering, 2023, doi:10.3390/bioengineering10121364_

Round 1

Reviewer 1 Report

Comments and Suggestions for Authors

This review is interesting for bioengineering and medicine community and could be accepted for publication. The topic is up to date and actual. Osteoporosis, marked by low bone mineral density and higher fracture risk, is a major health issue. Recent progress in medical imaging, especially CT scans, offers new ways for diagnosing and assessing osteoporosis. This review examines the use of AI analysis of CT scans to classify osteoporosis. By summarizing the relevant studies, authors aim to assess the effectiveness, constraints, and potential impact of AI-based osteoporosis classification via CT. A systematic search of electronic databases (PubMed, MEDLINE, Web of Science, clinicaltrials.gov) was conducted according to the Preferred Reporting Items for Systematic Reviews and Meta-Analyses (PRISMA) guidelines. Important considerations and limitations are also

discussed. While additional research is necessary to validate the clinical efficacy and reproducibility of these AI tools before incorporating them into routine clinical practice, these studies demonstrate the promising potential of using CT to opportunistically predict and classify osteoporosis without the need of DEXA. The subject addressed in this article is worthy of investigation. The information presented is new. The conclusions supported by the data. The manuscript is good illustrated and interesting to read. May be only some more detailed perspectives about the future research could be formulated in conclusions.

Comments on the Quality of English Language

Minor polishing.

Author Response

For Review Article: Artificial Intelligence Applications for Osteoporosis Classification using Computed Tomography

Point by Point Responses

Reviewer 1:

This review is interesting for bioengineering and medicine community and could be accepted for publication. The topic is up to date and actual. Osteoporosis, marked by low bone mineral density and higher fracture risk, is a major health issue. Recent progress in medical imaging, especially CT scans, offers new ways for diagnosing and assessing osteoporosis. This review examines the use of AI analysis of CT scans to classify osteoporosis. By summarizing the relevant studies, authors aim to assess the effectiveness, constraints, and potential impact of AI-based osteoporosis classification via CT. A systematic search of electronic databases (PubMed, MEDLINE, Web of Science, clinicaltrials.gov) was conducted according to the Preferred Reporting Items for Systematic Reviews and Meta-Analyses (PRISMA) guidelines. Important considerations and limitations are also discussed. While additional research is necessary to validate the clinical efficacy and reproducibility of these AI tools before incorporating them into routine clinical practice, these studies demonstrate the promising potential of using CT to opportunistically predict and classify osteoporosis without the need of DEXA. The subject addressed in this article is worthy of investigation. The information presented is new. The conclusions supported by the data. The manuscript is good illustrated and interesting to read. May be only some more detailed perspectives about the future research could be formulated in conclusions.

R1.1: Thank you for the kind comments. We have added more detailed perspectives about future research in our conclusion. Thank you for reviewing our manuscript.

Reviewer 2 Report

Comments and Suggestions for Authors

The authors used bioinformatics approaches for osteoporosis (OP) screening using CT images performed for other indications.

Comments

1.      The authors should define terms OP “classification”, “screening”, and “diagnostic criteria” and to make their mind of what they tried to study. As OP is classified based on etiology and severity of the disease, therefore, OP may be classified into primary and secondary types or can be classified as osteopenia, OP and severe OP, the authors definitely studied something else but OP classification. This should be corrected. 

2.      The authors should compare the efficacy of screening using AI applications for CT images performed for other indications and CT images performed exactly for OP diagnostics including clinical characteristics of the examined patients and considering diagnostic criteria according to National OP Foundation guidelines. This should be corrected.

3.      Simple summary versus Introduction: The authors should explain the contradiction between “reduced radiation” burden from CT scans and “high radiation dose” from DECT.

4.      All abbreviations should be disclosed on first use. All typos should be corrected.

5.      Section 3.1. (the end): The discrepancy in the highest AUC values should be corrected.

6.      Reference for Fig3 should be included to the text of the paper.

7.      Conclusion does not have sense as subject of the study is obscure.

Author Response

For Review Article: Artificial Intelligence Applications for Osteoporosis Classification using Computed Tomography

Point by Point Responses

Reviewer 2:

- The authors should define terms OP “classification”, “screening”, and “diagnostic criteria” and to make their mind of what they tried to study. As OP is classified based on etiology and severity of the disease, therefore, OP may be classified into primary and secondary types or can be classified as osteopenia, OP and severe OP, the authors definitely studied something else but OP classification. This should be corrected.  

R2.1: Our apologies for the confusion. In our study, we classified osteoporosis based on its severity (normal BMD, osteopenia and osteoporosis). We agree that there might be some confusion with regards to the definition of osteoporosis classification. We have added several lines at the end of the introduction as well as throughout the main text to emphasise the type of osteoporosis classification in our paper (severity instead of aetiology).

- The authors should compare the efficacy of screening using AI applications for CT images performed for other indications and CT images performed exactly for OP diagnostics including clinical characteristics of the examined patients and considering diagnostic criteria according to National OP Foundation guidelines. This should be corrected. 

R2.2: Thank you for your insights. Following your suggestions, we conducted an extensive literature search. To date, no studies have directly compared AI applications for CT images performed for purposes other than osteoporosis (e.g., CT scans of the abdomen and pelvis) and those specifically acquired for osteoporosis diagnosis. However, it is possible that the latter may yield greater accuracy due to the use of specific, rigorous protocols which optimise image acquisition and analysis. Recognizing the significance of this aspect, our study aims to pave the way for AI's opportunistic screening capabilities using CT scans undertaken for medical indications beyond osteoporosis. We have incorporated this consideration into our discussion. Additionally, our exploration of technical considerations, encompassing scan factors incluidng area sampling and contrast versus non-contrast images, offers valuable insights for future AI model development and research endeavors.

- Simple summary versus Introduction: The authors should explain the contradiction between “reduced radiation” burden from CT scans and “high radiation dose” from DECT.

R2.3: Thank you for your comments. We have included more details as to why there is reduced radiation burden on conventional CT relative to DECT (single energy versus dual energy acquisitions). We have also explained at the end of introduction how opportunistic screening can lead to overall reduction in radiation dose (reduced need for further radiological investigations for the diagnosis of osteoporosis). 

- All abbreviations should be disclosed on first use. All typos should be corrected.

R2.4: These have been corrected, including abbreviations.

- Section 3.1. (the end): The discrepancy in the highest AUC values should be corrected.

R2.5: This has been corrected. Thank you for the feedback.

- Reference for Fig3 should be included to the text of the paper.

R2.6: We have included a paragraph to illustrate Figure 3. Thank you for your suggestions.

- Conclusion does not have sense as subject of the study is obscure.

R2.7: Thank you for your valuable feedback. We have enhanced the conclusion of our study by providing additional context on the implications drawn from our findings. This includes providing the relevance of our study to the current literature and its potential for guiding future research. We have also outlined the limitations encountered in our review, shedding light on topics that need further exploration. We believe that this conclusion strengthens the overall contribution of our paper.

Reviewer 3 Report

Comments and Suggestions for Authors

Thank you for the opportunity to review this paper.

However I am afraid to say that I cannot recommend this paper for publication at this stage.

The paper is written well but lacks too many evidences as a Systematic Review needs.

Please find the comments below

Introduction

1.     The introduction lacks specific citations for the statements made regarding the prevalence of osteoporosis, its health implications, and the limitations of DEXA.

2.     The introduction could benefit from a more organized structure. It jumps between topics such as osteoporosis definition, diagnostic methods, and the potential of CT scans and AI, making it somewhat disjointed. Consider reorganizing the content for a more coherent flow.

3.     The section discussing the use of CT scans and dual-energy CT (DECT) for bone evaluation could be more explicit in explaining how these imaging techniques work and their relevance to osteoporosis diagnosis.

4.     The mention of DECT being used to assess local changes in BMD is introduced but not fully explained.

5.     The section discussing the potential future innovation of exclusively using CT scans for BMD measurement makes several assumptions. These assumptions, such as decreased expenses and radiation exposure, should be supported by evidence or discussed more cautiously, considering potential challenges.

6.     While the introduction outlines the potential benefits of AI in osteoporosis classification, it does not clearly identify the specific research gap or problem that this systematic review aims to address.

Methodology

1.     The methodology section briefly mentions the use of keywords and MeSH terms for the literature search but lacks details on how these terms were combined, any search filters or Boolean operators used, and whether any language or date restrictions were applied.

2.     While the methodology provides some information on the inclusion criteria, it is somewhat vague. It states that studies should use radiomics techniques, AI, or deep learning, but it would be beneficial to specify the minimum quality or methodological requirements for these techniques. Additionally, there is a lack of clarity regarding the comparison to conventional DXA studies, which should be elaborated on.

3.     The methodology section briefly mentions the use of a spreadsheet for data extraction but lacks details on the specific variables or data points that were extracted from each selected study.

4.     The methodology does not discuss any efforts made to identify or address potential publication bias, measures taken to ensure inter-reviewer agreement or how discrepancies between reviewers were resolved.

5.     The methodology section does not mention whether the selected studies were assessed for methodological quality or risk of bias, which is a crucial step in systematic reviews.

Results

1.     The Results section mentions that most studies lacked detailed data to create 2 × 2 contingency tables. This indicates a significant limitation in the analysis, as the absence of such data can affect the validity and robustness of the findings.

2.     The section provides summary statistics such as accuracy, sensitivity, specificity, and AUC for AI in classifying osteoporosis but does not offer a comprehensive view of the results.

3.     The paper mentions that studies with two-label classification achieved relatively higher performance compared to those with three-label classification. Still, it fails to provide a more in-depth comparative analysis of these two approaches.

4.     While the paper briefly mentions correlation values for different regions analyzed, it does not delve into the interpretation of these correlations.

5.     The Results section provides accuracy metrics for various studies but lacks a more in-depth discussion of the clinical implications of the diagnostic accuracy achieved by AI in classifying osteoporosis.

Discussion

1.     The discussion primarily focuses on highlighting the advantages of using AI and CT scans for osteoporosis detection, but it lacks a critical analysis of potential limitations and challenges.

2.     While the paper mentions the advantages of AI-driven methods, it doesn't provide a comprehensive comparison with existing methods, such as DEXA scans.

3.     Some of the comparisons between different AI models and techniques are presented without information about statistical significance.

4.     The paper relies heavily on a few studies to make its points.

5.     The discussion does not address the challenges related to the reproducibility of AI models in different settings and populations.

6.     The discussion contains technical terms and acronyms like "AUC," "BTMs," and "radiomics" without clear explanations.

7.     The discussion briefly mentions potential future applications related to molecular and genetic biomarkers but does not delve deeper into the challenges and implications of these applications.

Author Response

For Review Article: Artificial Intelligence Applications for Osteoporosis Classification using Computed Tomography

Point by Point Responses

Reviewer 3:

Introduction

  1. The introduction lacks specific citations for the statements made regarding the prevalence of osteoporosis, its health implications, and the limitations of DEXA.

R3.1: Thank you for your suggestion. We have included two sentences on the prevalence of osteoporosis. We have also expanded on the health implications of osteoporosis, including the occurrence of fragility fractures and their impact on the patient and healthcare system.

There is also now a short paragraph on the limitations of DEXA “Firstly, DEXA's diagnostic capability is mainly confined to BMD alone (30, 31) and suboptimal screening rates have been reported (32, 33). Additionally, it is constrained by its planar technique (2-Dimensional measurement) for assessing and quantifying BMD and predicting fracture risk (34, 35). Moreover, DEXA measurements are sensitive to degenerative changes, leading to potential overestimation of BMD (36-38), and the presence of overlying structures (such as atherosclerosis (39)) or morphological abnormalities (post-laminectomy, metallic implants etc. (40-42)) may also impact BMD measurements.” Thank you for your valuable input.

  1. The introduction could benefit from a more organized structure. It jumps between topics such as osteoporosis definition, diagnostic methods, and the potential of CT scans and AI, making it somewhat disjointed. Consider reorganizing the content for a more coherent flow.

R3.2: Thank you for your suggestions. We have restructured the paragraphs to achieve a more coherent flow in the introduction. The sequence is as follows: (1) Introduction to osteoporosis, (2) Conventional diagnosis using DEXA, (3) Limitations of DEXA, (4) an exploration of an alternative approach, specifically DECT, (5) the identified limitations associated with DECT/CT, (6) the potential applications of CT, and (7) the introduction of AI and its potential role in addressing these challenges. We believe that these changes contribute to a clearer and more cohesive narrative and thank you for your feedback.

  1. The section discussing the use of CT scans and dual-energy CT (DECT) for bone evaluation could be more explicit in explaining how these imaging techniques work and their relevance to osteoporosis diagnosis.

R3.3: We have added a short paragraph on how DECT is used to evaluate and measure bone mineral density, thereby diagnosing osteoporosis.

  1. The mention of DECT being used to assess local changes in BMD is introduced but not fully explained.

R3.4: We have added a short paragraph detailing how DECT can evaluate local changes, specifically its efficacy in accurately distinguishing trabecular from cortical bone. This approach not only enhances precision but also offers valuable insights into local parameters such as trabecular spacing and connectivity, which are instrumental for bone health evaluation.

  1. The section discussing the potential future innovation of exclusively using CT scans for BMD measurement makes several assumptions. These assumptions, such as decreased expenses and radiation exposure, should be supported by evidence or discussed more cautiously, considering potential challenges.

R3.5: Thank you for your suggestions. We have provided a few references to support these statements. In particular, for decreased expenses in Cheng et al. (70), it was stated that the average price for a DXA examination was 15.7 US dollars (USD) (110 Chinese Yuan Renminbi [RMB]) versus 17.2 USD (120.5 RMB) for QCT. Furthermore, if CT performed for other reasons (e.g., abdominal pain) is able to diagnose osteoporosis without the need for additional DEXA, the patient can potentially save on costs for additional investigations, along with reducing radiation exposure. This was also supported by three other references (71-73).

  1. While the introduction outlines the potential benefits of AI in osteoporosis classification, it does not clearly identify the specific research gap or problem that this systematic review aims to address.

R3.6: Thank you for the suggestion. We have included a short paragraph at the end of the introduction, stating the current research gap and how our study aims to bridge this gap.

Methodology

  1. The methodology section briefly mentions the use of keywords and MeSH terms for the literature search but lacks details on how these terms were combined, any search filters or Boolean operators used, and whether any language or date restrictions were applied.

R3.7: Thank you for your suggestion. We have provided details on the use of Boolean operators in the following (“Artificial intelligence” OR “AI” OR “deep learning” OR “machine learning” OR “convolutional neural network*” OR “neural network” OR “radiomics”) AND (“osteoporosis” OR “osteopenia” OR “osteopaenia” OR (“bone” AND “mineral” AND “density”) OR “BMD”) AND (“DEXA” OR “absorptiometry”) AND (“CT” OR (“Computed” AND “Tomography”)). In terms of search restrictions, we have also included these in the inclusion and exclusion criteria, with a date restriction identified as the 14 Aug 2023.

  1. While the methodology provides some information on the inclusion criteria, it is somewhat vague. It states that studies should use radiomics techniques, AI, or deep learning, but it would be beneficial to specify the minimum quality or methodological requirements for these techniques. Additionally, there is a lack of clarity regarding the comparison to conventional DXA studies, which should be elaborated on.

R3.8: Dear reviewer, thank you for your comments. The studies we included utilise various machine learning/deep learning methods to classify osteoporosis, and their performance is compared with the gold standard methods of osteoporosis diagnosis and classification using DEXA. We have included these details in section 2.2 under the study screening and selection criteria.

  1. The methodology section briefly mentions the use of a spreadsheet for data extraction but lacks details on the specific variables or data points that were extracted from each selected study.

R3.9: Thank you for the suggestions. We have included this portion under the Data Extraction and Reporting segment (2.3). Details on the specific variables and data points extracted include (1) Research article details, (2) Main clinical use, (3) Research study details, (4) Machine Learning techniques and (5) Performance.

  1. The methodology does nt discuss any efforts made to identify or address potential publication bias, measures taken to ensure inter-reviewer agreement or how discrepancies between reviewers were resolved.

R3.10: Thank you and apologies for the oversight. The review was conducted independently by two authors, and any discrepencies between reviewers were resolved by a third author. We have included this portion in the methodology segment.

  1. The methodology section does not mention whether the selected studies were assessed for methodological quality or risk of bias, which is a crucial step in systematic reviews.

R3.11: Regarding our approach to analysis, we have discussed this among the authors and opted for a scoping literature review using a systematic method rather than a systematic review. This decision is rooted in the predominantly retrospective nature of the included studies, presenting results in various formats. We acknowledge the limitations of this review style and have addressed them in the paper's conclusion. Additionally, we have adjusted the title of the paper to accurately reflect our research methods. We appreciate your valuable insights and comments.

Results

  1. The Results section mentions that most studies lacked detailed data to create 2 × 2 contingency tables. This indicates a significant limitation in the analysis, as the absence of such data can affect the validity and robustness of the findings.

R3.12: Thank you for your comments. Regrettably, due to the limitations of the existing literature, we are unable to conduct a substantive statistical analysis, which is typically expected in a systematic review. This is an inherent limitation of this article, and we do acknowledge it with a short paragraph in the conclusion. Please also see our comments for R3.11.

  1. The section provides summary statistics such as accuracy, sensitivity, specificity, and AUC for AI in classifying osteoporosis but does not offer a comprehensive view of the results.

R3.13: We appreciate the reviewer's feedback and acknowledge their concern regarding the comprehensiveness of the results section in our study on AI for classifying osteoporosis on CT. The provided summary statistics, including accuracy, sensitivity, specificity, and AUC, were intended as a concise representation of the primary performance metrics of the included AI models. We believe we have provided the most comprehensive review of the results possible given the heterogeneity in the study designs and metrics provided. This has been acknowledged as a limitation in the conclusion.

  1. The paper mentions that studies with two-label classification achieved relatively higher performance compared to those with three-label classification. Still, it fails to provide a more in-depth comparative analysis of these two approaches.

R3.14: Thank you for the feedback regarding the comparative analysis of two-label and three-label classification in our paper. You have raised a valid point, and we acknowledge that a more in-depth comparative analysis between these two approaches could provide valuable insights. We have included a few sentences in the results section elaborating on why two-label classification can achieve higher accuracy.

  1. While the paper briefly mentions correlation values for different regions analyzed, it does not delve into the interpretation of these correlations.

R3.15: Thank you for the insights. We noted that investigations concentrating on the cervical and thoracic vertebra achieved only a moderate degree of correlation (r = 0.270 to 0.670) in contrast to studies encompassing the lumber vertebra (r = 0.582-0.911). We delve into this topic in detail in our discussion with a dedicated section (4.2.3. technical considerations: areas sampled).

  1. The Results section provides accuracy metrics for various studies but lacks a more in-depth discussion of the clinical implications of the diagnostic accuracy achieved by AI in classifying osteoporosis.

R3.16: Dear reviewer, thank you for your comments. We have elaborated on the clinical implications of the diagnostic accuracy achieved by AI in classifying osteoporosis in the discussion section, specifically in section 4.1 Advantages and Efficacy. 

Discussion

  1. The discussion primarily focuses on highlighting the advantages of using AI and CT scans for osteoporosis detection, but it lacks a critical analysis of potential limitations and challenges.

R3.17: Dear reviewer, thank you for the comments. We have incorporated sections addressing the limitations in the discussion and conclusion. We have also delved into the potential challenges and constraints associated with deploying AI models for osteoporosis classification using CT images.

  1. While the paper mentions the advantages of AI-driven methods, it doesn't provide a comprehensive comparison with existing methods, such as DEXA scans.

R3.18: Dear reviewer, thank you for your comments. While there are existing studies in the literature exploring the application of AI on DEXA scans or DECT to enhance osteoporosis diagnosis, we would like to clarify that our review article specifically centers on the feasibility of employing AI methods for osteoporosis classification using CT images. The primary focus is on the potential utility of such methods for opportunistic screening, considering the added benefits they may bring. Despite this, we have tried to compare with DEXA as far as possible. Future research will hopefully provide more insights into the comparison of AI techniques on CT with existing methods. Thank you for your understanding and valuable comments.

  1. Some of the comparisons between different AI models and techniques are presented without information about statistical significance.

R3.19: Thank you for the insights on the comparisons between various AI models and techniques in our study. We recognize the importance of statistical significance in making meaningful comparisons. Yet, given the diverse nature of the included studies and significant variations in methodologies, conducting a direct analysis to assess statistical significance has posed challenges. As a result, this study primarily serves as a concise literature review with a systematic approach. Furthermore, making direct comparisons between different AI models and techniques is challenging, considering the varied use of distinct AI models and methods across the different papers. The use of more established criteria for AI research in the future should make this comparison feasible. We have mentioned these issues as limitations in the conclusion. We appreciate your valuable feedback.

  1. The paper relies heavily on a few studies to make its points.

R3.20: Thank you for your comment. We acknowledge that our review draws from a limited number of studies. This limitation stems from the novelty of the topic (AI models have only recently become feasible with improvements in computing resources), coupled with significant variations in study designs, such as areas sampled and the types of AI models employed across different studies. Recognizing this as a limitation, we have openly addressed this in the conclusion, emphasizing the need for future studies to address these challenges.

  1. The discussion does not address the challenges related to the reproducibility of AI models in different settings and populations.

R3.21: Dear reviewer, thank you for your comments. We have added a segment on the challenges in implementation in the discussion section and provided comments on the reproducibility in the conclusion. Thank you for your suggestions.

  1. The discussion contains technical terms and acronyms like "AUC," "BTMs," and "radiomics" without clear explanations.

R3.22: Dear reviewer, thank you for your comments. We have defined the term radiomics in the introduction. We have also defined the abbreviations AUC in table 2 as well as in the results segment. We have added elaborations on BTM, specifically bone turnover peptides, which is a type of peptide produced during bone turnover, which can be used to evaluate osteoporosis at the molecular level. We have modified the paragraph to answer your comments. Thank you.   

  1. The discussion briefly mentions potential future applications related to molecular and genetic biomarkers but does not delve deeper into the challenges and implications of these applications.

R3.23: Dear reviewer, the exploration of future AI applications associated with molecular and genetic biomarkers in osteoporosis is a novel area that remains largely unexplored. In our manuscript, we have provided a brief explanation of the exciting potential implications of such applications. However, addressing the challenges of implementation may extend beyond the current scope of our study, and really requires further research to establish a more comprehensive understanding. We will continue this exploration in our future projects. Thank you once again for your valuable comments.

Round 2

Reviewer 2 Report

Comments and Suggestions for Authors

After revision, the review became better shaped.

Comments

1.     “DECT has shown near similar sensitivity to gold standard DEXA…” The level of sensitivity similarity should be indicated.

2.      The first indication on term “HU value” in the Introduction should be disclosed.

3.     Discussion: FRAX method should be defined here.

Author Response

For Review Article: Artificial Intelligence Applications for Osteoporosis Classification using Computed Tomography

Point by Point Responses

Reviewer 2:

- “DECT has shown near similar sensitivity to gold standard DEXA…” The level of sensitivity similarity should be indicated.

R2.1: Thank you for your comments. We agree that we should elaborate slightly more on this statement. We have included some percentages, with specific studies quoted by Booz et al. and Gruenewald et al. later in the sentence. By including specific percentage ranges or values, we hope this would provide a clearer understanding of the extent to which DECT and DEXA exhibit similar sensitivity.

- The first indication on term “HU value” in the Introduction should be disclosed.

R2.2: Thank you for your comments. We have disclosed the term in the introduction as per your advice.

- Discussion: FRAX method should be defined here.

R2.3: Thank you for your comments. We have defined the FRAX method as per your advice.

Reviewer 3 Report

Comments and Suggestions for Authors

Congratulations to the authors 

The paper is now highly improved and an asset to the readers

All the best

Author Response

For Review Article: Artificial Intelligence Applications for Osteoporosis Classification using Computed Tomography

Point by Point Responses

Thank you very much for taking the time to review this manuscript.

We appreciate your help and valuable inputs in improving our papers.

Best Regards
